# Characterization of alveolar epithelial cells type II during postnatal lung development in relation to alveolarization – Stereological studies of rat lungs

Julia Hüttmann[1], Lars Knudsen[1,2], Andreas Schmiedl [1,2]*

1 Institute of Functional and Applied Anatomy, Hannover Medical School, Hannover, Germany,
2 Biomedical Research in Endstage and Obstructive Lung Disease Hannover (BREATH), Member of the German Center for Lung Research (DZL), Hannover, Germany

* Schmiedl.Andreas@mh-hannover.de

## Abstract

### Objective

Rats are born with morphologically immature lungs, but intact surfactant system. The aim of this study was to characterize the surfactant producing alveolar epithelial cells type II (AEII) during alveolarization and find relationships between the intracellular surfactant pool and alveolar surface area, lung volume and body weight.

### Methods

After exsanguination, lungs of 3, 7, 14, 21 and 90 days old rats were inflated with a pressure of 10 mm $H_2O$ and fixed by perfusion and prepared for light and electron microscopy. Using different stereological parameters AEII were characterized.

### Results

At day 21, the end of bulk alveolarization, the alveolar surface and the number of AEII increased significantly but their volume and size did not change compared to values before alveolarization. The number of AEII, but not the AEII volume correlated significantly with alveolar surface and lung volume. The size and volume weighted mean volume of lamellar bodies (Lb) as well as the Lb volume per AEII did not change during alveolarization. Total Lb volume was significantly higher at the end of bulk alveolarization compared to values before alveolarization.

### Conclusion

The adaptation of the intracellular surfactant during postnatal development occurred predominantly by increasing the number of AEII.

**Data availability statement:** All relevant data are within the manuscript.

**Funding:** The author(s) received no specific funding for this work.

**Competing interests:** The authors have declared that no competing interests exist.

## Introduction

The alveolar epithelial cells type II (AEII) synthesize, store, secrete and reuptake the surface active agent (surfactant), which is necessary to reduce the surface tension at the gas-liquid boundary of the distal airspaces and to guaranty efficient breathing [1–3]. Furthermore, it helps to prevent fluid influx from the septal capillaries [4,5] and has immunomodulatory functions [6]. The surfactant, a mixture of predominantly phospholipids and proteins, consists of an intracellular part and intraalveolar part. The intracellular surfactant includes lamellar bodies (Lb), the storage organelles of surfactant, multivesicular bodies (mvb), and composite bodies (cb) [7]. The intraalveolar surfactant comprises the surfactant film, spread at the air-liquid boundary of alveoli and active precursors and inactive degradation products lying underneath [4,8,9]. All surfactant proteins (SP) are synthesized on the rER, modified in the Golgi apparatus and mvb. The hydrophilic SP-A and SP-D are secreted predominantly continuously via mvb per exocytosis [10,11], the hydrophobic SP-B and SP-C are taken up in immature Lb by fusing with mvb to cb, so that mature Lb containing SP-B and SP-C are secreted per exocytosis [12,13]. The SP-A has predominantly immunomodulatory functions [14], SP-D has an immunomodulatory effect and is necessary for the surfactant homeostasis [15]. SP-B and SP-C are necessary for constitution, stabilization and purification of the surfactant layer [16,17].

During morphological lung development epithelial differentiation into AEII occurred in the so called canalicular phase between gestation weeks 16–26 in humans and between gestation days 18.5–20 in rats [18]. In this period the Lb also form and accumulate as intracellular lamellar inclusions [19–21]. Furthermore, in this fetal period, the first structural properties of the air blood barrier are visible and minimal surfactant production is found [21–23]. The canalicular period is followed by the saccular phase. The parenchyme consists of thin-walled channels and thick-walled saccules. The saccules are surrounded by thick primary septa containing a double-layered capillary bed. The saccular phase ends in humans already before birth and alveolarization begins prenatally [18,22]. In contrast to human lung, the rat lung is still in the saccular phase at birth and therefore morphologically immature at birth comparable with premature infants with a gestational age between the 24th and 27th week [23,24]. This saccular phase extends until the third postnatal day and alveolarization starts at postnatal day 4 [18]. Based on qualitative and some morphometric evaluations, postnatal lung development in rats can be divided in three phases: before alveolarization (day 1–3), classical or bulk alveolarization with vascular maturation (days 4–21) and continued alveolarization (day 21 – day 60, young adulthood) [18,22,25]. Therefore, rats as well as mice are suitable for studying perinatal morphological lung development under normal conditions [25–27]. Furthermore, rodents are frequently used as animal models for studying diseases of premature infants with a gestational age between 24 and 27 weeks such as bronchopulmonal dysplasia (BPD) [28,29]. Different authors, using different designed based stereological methods verified the parenchymatous changes in lungs of healthy pups [25,30–36]. Using the disector method, an increase of alveolar number during the bulk alveolarization period was found in rat and mouse pups [25,33,35], as well as in humans between the 32nd gestational week and two

years after birth [37,38]. As shown in rats, alveolarization decelerated after postnatal day 21, but continued until adulthood (continued alveolarization) [25]. Functional lung development means the maturation of the surfactant system and therefore the development of the surfactant producing AEII and their surfactant storing Lb [22]. Normal breathing and prevention of alveolar collapse during expiration is only possible if the surfactant system is morphologically and functionally intact already after birth [39,40]. Earlier studies have shown that the expression of the immunomodulatory SP-A, and SP-B is differentially associated with morphological lung maturation and correlates with increased septation of alveoli as indirect clue for alveolarization [41]. The differentiation of AEII as well as the occurrence of surfactant in lungs was verified already during prenatal development [42–44].

During lung development, the alveolar surface increases as well as the lung volume and body weight as already shown by others [25,45,46]. Some publications described the intracellular surfactant, especially the Lb during postnatal development too [20,47–50].

### The aim of the study

Although rats are born with morphological immature lungs, but an intact surfactant system, there is only little knowledge about the relationships between the morphological components of the surfactant system and the degree of alveolarization. It is expected that during lung development the number and volume of AEII and their surfactant storing organelles will also increase, but systematic investigations about the fate of these cells in relation to alveolarization and lung volume during postnatal development are missing.

Therefore, this study was done, to answer following question using different designed based stereological methods: Is there a relationship between growth of AEII and their surfactant storing organelles to alveolar surface and lung volume before, during and after the end of bulk alveolarization in comparison to adults?

## Materials and methods

### Ethic statement

F344 rats of either sex were kept under specific pathogen- and germ-free conditions and had ad libitum access to food and water. The study was carried out in strict accordance with the recommendations in NIH Guidelines for the Care and Use of Laboratory Animals [NIH Publication No. 85. reprint 2002]. The study protocol was approved by the ethics board State Office of Lower Saxony for Consumer Protection and Food Safety (Niedersächsisches Landesamt für Verbraucherschutz und Lebensmittelsicherheit, LAVES) The approval number is 2015/87, Oldenburg, Germany).

### Lung fixation

Lungs from 3-day (n = 7), 1-week (n = 6), 2-week (n = 6), and 3-week (n = 6) old as well as adult Fischer rats (n = 6) are perfusion-fixed in situ via the pulmonary artery and then immersion-fixed. For this, rats of different age were anesthetized with isoflurane under a glass bell. Deep anesthesia was checked by complete lack of a response to the pedal reflex pinch. Under deep anesthesia, the animal's abdominal cavity was opened and the abdominal aorta was immediately incised to exsanguinate the animal leading to the animal´s death without any suffering. Afterwards, a cannula was inserted via the larynx, advanced into the trachea, tied and connected with a u-shaped hydrostatic pressure column. After thoracotomy, the right ventricle was cannulated and the lungs were flushed with NaCl/heparin solution via the pulmonary artery. The left atrial appendage was cut to relieve the volume pressure. At the end of preperfusion, the lungs were inflated up to 30 cm $H_2O$, deflated to 10 cm $H_2O$ and then fixed by vascular perfusion with 1.5% paraformaldehyde and 1.5% glutaraldehyde in 0.15M HEPES buffer. Afterwards the heart-lung block was removed and was fixed by immersion for at least 24h [41].

Before sampling and further processing the lung volume was determined by the Archimedes principle [51]. After volume measurements, the lungs were embedded in 2% aqueous agar, and each organ was cut from apical to caudal into 1 mm

slices using a tissue slicer. Tissue slices of the right lung were taken for light and electron microscopy starting alternately with a random number. Tissue slices collected for electron microscopy were additionally cut into tissue blocks with a size of 1–2 mm³. After additional immersion fixation, specimens were rinsed repeatedly in HEPES buffer, then in cacodylate buffer. Postfixation followed in 1% $OsO_4$ in 0.1 M cacodylate buffer for 2 h. Again, specimens were rinsed in cacodylate buffer then in distilled water and stained en bloc overnight at 4–8°C in half-saturated aqueous uranyl acetate solution. After dehydration in an ascending acetone series embedding followed completely in glycol methacrylate (Technovit 8100, Heraeus Kulzer GmbH, Hanau, Germany) or in epoxy resin, 1.5 µm methylacrylate and 1 µm epon sections were cut and stained with toluidine blue. Finally, ultra-thin sections (70 nm) were stained with lead citrate and uranyl acetate, and examined with a electron microscope (Morgagni 268, FEI, Eindhoven, The Netherlands).

## Stereology

The stereological evaluation was carried out using the point and intersection counting according to the guidelines for quantitative assessment of lung structure [52]. To get information about morphological lung maturation, the surface density of alveoli ($S_V$(alveoli/par)) as an indirect parameter for alveolarization was determined light microscopically using the software visiopharm (Visiopharm, Hoersholm, Denmark). Test fields were sampled all over the section by the software according to the systematic random area sampling to guarantee that each area of the section has the same chance to be evaluated [41]. To get values independent of the reference space surface and volume densities were multiplied with $V_V$(par/par + nonpar), the volume fraction of lung parenchyma within the lung, and lung volume.

The number and volume of AEII were determined using the physical disector [53]. Accordingly, the first and fourth section of a consecutive row of toluidine blue stained 1µm thin sections of epoxy resin embedded samples, that were mounted on the same glass slide, were used as a disector pair. The disector height was 3 µm. The number of AEII and the number-weighted mean volume of AEII per lung were determined as described earlier [54]. With the evaluated disector data at first the numerical density (Nv(AEII/ par)) was determined:

$$NV\ (AEII\ /\ par)\ =\ 1/2\ \times a\ (f\ )\ \times\ h\ \times\ Q^-AEII\ /\ frames\ /\ par$$

The number of AEII per lung: N (AEII, lung) = $N_V$ (AEII/ par) x V (lung)
h = disector height, a(f) = area per test frame, a(p) = area per test point, Q- = counting events

The number-weighted mean volume (vn(NAEII)) was determined during counting of AEII using the nucleator method. Therefore, the volume of the disector counted cells were determined by measuring the distance from a random point placed on the cell nucleus to the cell membrane in a random direction and at defined angles to this direction [55].

Further parameters that characterize possible changes in the AEII are determined on the electron microscopic level. Alveolar epithelial cells type II (AEII) were collected according to the systematic random sampling [56] using a transmission electron microscope (TEM) (Morgagni II 268, Fa. FFEI, Oregon USA) provided with a digital camera (Veleta CCD, Olympus SIS, Münster, Germany). Stereological evaluation was performed according to the point and intersection counting method using the software Stepanizer1stereology tool Version 1 [57]. We determined the volume densities $V_V$ of subcellular compartments (multivesicular bodies (mvb), composite bodies (cb), nuclei (nc) mitochondria (mi), cytoplasm (cp) and lamellar bodies (Lb) as described earlier [58]. Also the surface density of Lb ($S_V$ (Lb/AEII) and the volume to surface ratio of AEII and Lb ($V_S$-ratioAEII, $V_S$-ratio Lb) were determined. The volume weighted mean volume of Lb (vV(Lb,AEII)) was determined by means of the point sampled intercepts as described earlier [47,54,58].

## Statistics

All values are given in mean ± SD. Because our values are not normally distributed, we used the nonparametric Kruskal-Wallis test to detect significant differences instead of the One-Way ANOVA test for normally distributed values. Multiple

comparisons were corrected with the post-hoc Dunn´s multiple comparison test, because we test different comparisons. Correlation analysis between different parameters were carried out. Correlation analyses were carried out using the non-parametric Spearman correlation test, which provided a coefficient of correlation (r) and a measure of statistical significance (p). A level of two tailed $p < 0.05$ was considered to be significant. The Graph Pad Prism 6.07 (Statcom, Witzenhausen, Germany) was used.

## Results

### Body weight and lung volume of rats

The body weight and lung volume did not change significantly during the first two postnatal weeks. A slight, but significant increase was found at the end of bulk alveolarization at postnatal day 21. Compared to 3 days old rats, body weight and lung volume increased already significantly at the end of bulk alveolarization (Table 1). There was a significant correlation between increase of lung volume and body mass not only during and after bulk alveolarization, and within all age groups (Table 2-3).

### Histology of lung parenchyma

In 3 day old pups different sized sacculi were found (Fig 1a). The more or less thick septa contained a double layered capillary bed (arrows). Enlarged airways were seen. After one postnatal week, signs of alveolarization were visible. Sporadically secondary septa are formed, visible first as protrusions reaching in the airspace sprouting from the primary septa (arrows). The septa contained also a double- layered capillary bed, but seemed locally smaller than in 3 d old rats (Fig 1b). At postnatal day (pnd) 14, advanced alveolarization was seen. In lung parenchyma predominantly secondary septa and more or less small alveoli were found (Fig 2a). Also, alveolar ducts could be differentiated (DA). The septa retained a double capillary bed (arrows). However, septa with single layered capillary bed occurred sporadically (arrow heads). After 21 postnatal days, the main part of alveolarization and capillary maturation was finished. Alveoli surrounded by septa with single layered capillary beds were visible (Fig 2b). In adults, predominantly signs of a fully developed mature lung parenchyme were seen (Fig 3).

### Characterization of the septal surface

The total alveolar or septal surface, as an indirect parameter for alveolarization, which was already published earlier with another animal number of 3 days old pups [41] showed that during the first 14 postnatal days the alveolar surface (S(alveoli, lung)) did not change. At the end of bulk alveolarization, however, a significant increase was determined. The significant highest values were found in adults (Fig 4a).

### Characterization of AEII

The absolute number of AEII (N(AEII, lung)) showed no differences before and during alveolarization until postnatal day 14. A significant increase was found on postnatal day 21 at the end of bulk alveolarization compared to values before and

**Table 1. Body mass and lung volume during postnatal development.**

|  | 3 days (d) old n=7 | 7 days old n=6 | 14 days old n=6 | 21 days old n=6 | 90 days old n=6 |
|---|---|---|---|---|---|
| Body Mass | 6.47±0.39 g | 12.61±0.34 g | 21.43±0.27 g+ | 30.38±0.41 g* | 194.80±55.96 g*.# |
| Lung Volume | 0.24±0.03 ml | 0.44±0.08 ml | 0.55±0.03 ml | 0.86±0.05 ml* | 3.16±0.65 ml*.# |

There is a signifcant difference between the different age groups (p<0.0001 (Kruskal-Wallis).

Post hoc test: significant differencens: * with p<0.05 compared to 3 d, # with p<0.05 compared to 7 d, +with p=0.13 compared to 3 d.

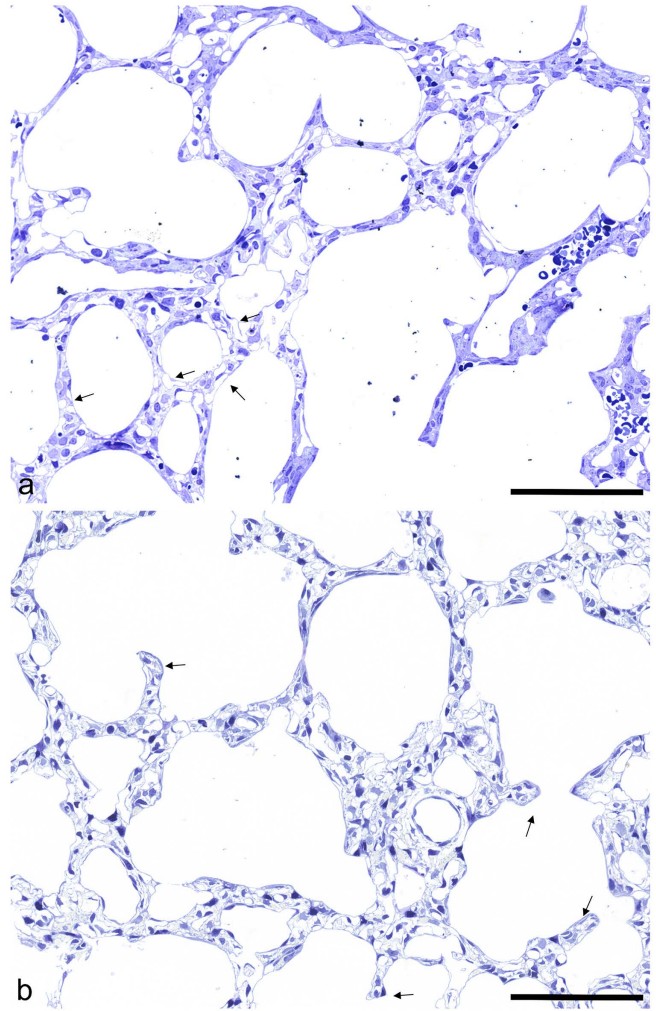

**Fig 1. Histology of lung parenchyma.** a) 3 days old rats, before alveolarization: saccules surrounded by thick primary septa containing a double layered capillary bed are (arrows) seen (scale bar; 200μm). b) 7 days old rats, during alveolarization: Protrusions of secondary septa are seen sporadically (arrows). The septa appear still thick (scale bar; 200μm).

during alveolarization. No significant differences between postnatal day 21 and 90 were seen. Highly significant differences in the AEII numbers were seen between 3 day old pups and adults (Fig 4b).

To get some information about the distribution of AEII on the alveolar surface during and after alveolarization, we determined the number of AEII per alveolar surface (N(AEII,lung)/S(alveoli, lung). Interestingly the highest values were evaluated before and after the first week of alveolarization (Fig 4c). Afterwards, the AEII number per surface decreased. The significant lowest values contained lungs of adult rats (Fig 4c). Thus, the number of AEII decreased with increased alveolar surface.

To get information about the relation between number of AEII and lung volume, we determined the quotient N(AEII, lung)/ lung volume (Fig 4d). The quotient did not change until postnatal day 14. In adults, significantly lower values were found compared to 3 and 7 day old rats. The highest values were determined at postnatal day 7 (Fig 4d).

Linear regression analyses between number of AEII and lung volume or alveolar surface exhibit a significant linear correlation during alveolarization (Table 2). Regarding all groups a significant relationship was found too (Table 3).

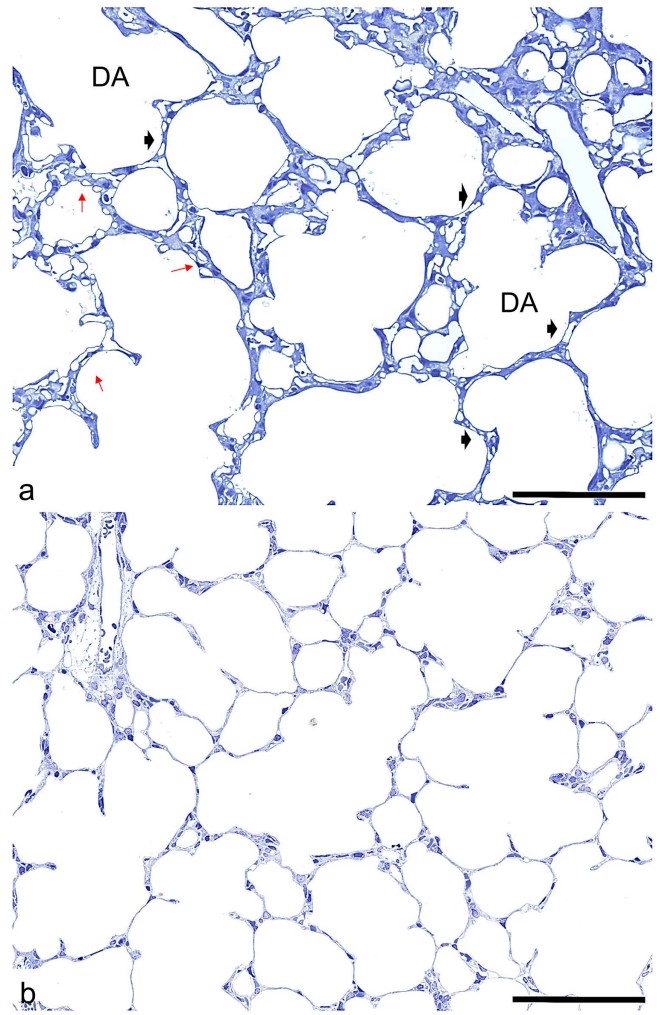

**Fig 2. Histology of lung parenchyma.** a) 14 days old rats. alveolarization is well advanced. Some alveoli and a lot of alveolar ducts (DA) are visible. Protrusions of secondary septa are often seen. Double (red arrows) and single layered (black arrow heads) capillary beds can be seen. (scale bar; 200µm). b) 21 days old rats: at the end of bulk alveolarization and microvascular maturation numerous alveoli with more or less small alveolar septa and alveolar ducts (DA) are seen. The capillary bed is single layered. The size of airspaces is smaller than in the younger developmental stages (scale bar; 200µm).

The number weighted mean volume (vN(AEII)) as a measure for AEII volume did not change during alveolarization. Values were significantly higher in adults compared to values before alveolarization and after the first week of alveolarization (Fig 5a). No correlation was determined between vN(AEII) and N(AEII, lung) before and during bulk alveolarization (Table 2). However, over all groups both parameters showed a significant correlation (Table 3)

The size of AEII determined as $V_s$-ratio AEII, a parameter, which is independent of the reference space, was comparable during the whole bulk alveolarization period. A significant increase was seen in adults compared to 7 and 14 days old rats (Fig 5b).

To answer the question in which way vN(AEII) and the $V_s$-ratioAEII related to the lung volume and the total alveolar surface before, during and after alveolarization, we formed the quotient of these parameters (Fig 5c-5f). We found that the vN(AEII, lung)/volume decreased significantly after 21 postnatal days and showed the significant lowest values in

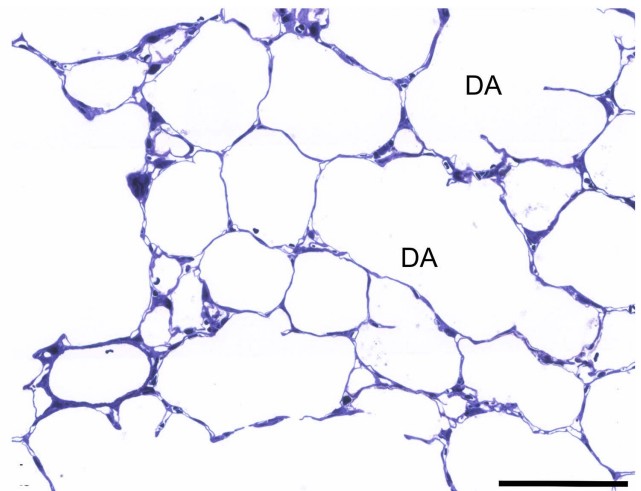

**Fig 3. Histology of lung parenchyma.** Adult rats: alveolar ducts (numerous small alveoli with thin alveolar septa and (scale bar; 200μm).

adults (Fig 5c). Thus, the lung volume has significantly more increased than the volume of AEII at the end of bulk alveolarization and in adults. The ratio vN(AEII)/S(alveoli, lung) showed the highest values before alveolarization and the highly significant lowest values in adulthood (Fig 5d). The alveolar surface increased during alveolarization considerably faster than cell volume increased. There was no correlation between vN(AEII) and alveolar surface during alveolarization, but a significant correlation over all age groups (Tables 2, 3).

The $V_S$-ratioAEII related to lung volume decreased. Significant differences compared to the 3 days old pups occurred firstly after the end of bulk alveolarization (Fig 5e). Compared to values before alveolarization highly significant values were obtained in adulthood. The data showed that changes in lung volume are greatly more pronounced than changes in AEII size during postnatal development. Correlation analyses exhibited only a significant correlation over all age groups (Table 2, 3).

The quotient $V_S$-ratioAEII/alveolar surface was highest before alveolarization and highly significant lowest in adults (Fig 5f). The alveolar surface increased considerably faster than AEII grow. Correlations were only found over all age groups (Table 2, 3).

### Characterization of the ultrastructure of the AEII

Looking at the ultrastructure of AEII before, during and after alveolarization in comparison to adults, no differences were visible qualitatively (Fig 6a-6e). The differently sized AEII with apical microvilli lied on a continuous basement membrane in corners or niches between capillaries at insertions of the alveolar septa independent of the investigated age. The subcellular organization in AEII was similar independent of the postnatal stage of development. The nuclei, when cut, exhibited small clumps of heterochromatin on the inner side of the nuclear envelope. In the cytoplasm, organelles such as mitochondria with densely packed cristae and dark matrix, ER, Golgi complex and vesicles of different origin were normally distributed. Lb, the storage organelles of surfactant, are visible as dense, ovoid membrane bound granules with closely spaced thin lamellae. Lb varied in size and were normally distributed in the cytoplasm in all age groups (Fig 6a-6e). Sporadically mvb and cb were found (Fig 6b, 6d). Interestingly, in contrast to adult rats, in lungs of young rats sporadically AEII with giant Lb in their cytoplasm were seen independent of the stage of morphological lung development (Fig 7a-7d).

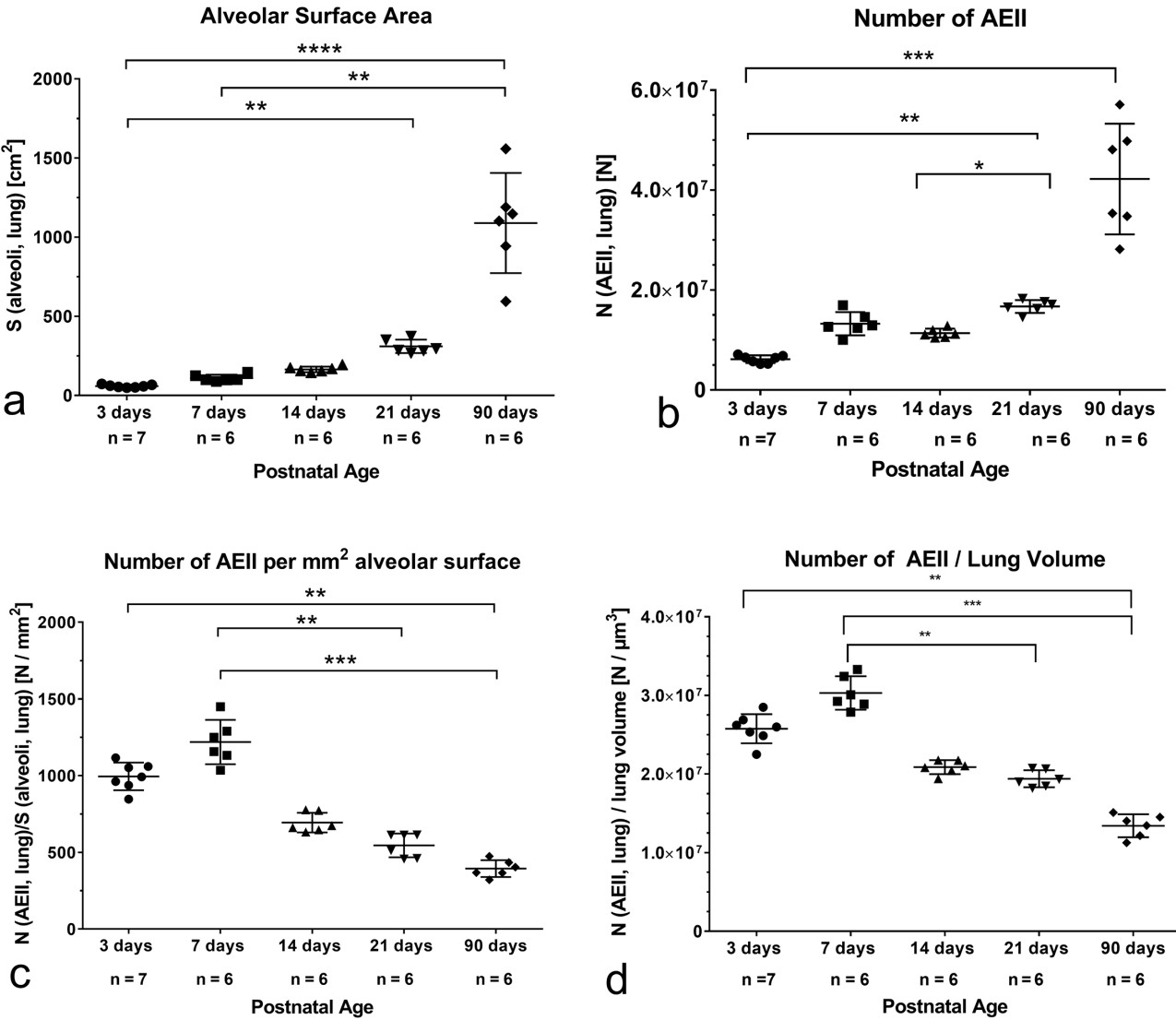

**Fig 4. Various stereological parameters characterizing lung parenchyma and AEII before, during and after alveolarization as well as in adults.** a) Compared to 3 days old rats a significant increase of the total alveolar surface (S(alveoli, lung)) was determined after the end of bulk alveolarization and in adults. This modified figure is reused from PLOS ONE, published first as Fig 3b by Roeder F, Knudsen L, Schmiedl A. The expression of the surfactant proteins SP-A and SP-B during postnatal alveolarization of the rat lung. PLoS One. 2024 Mar 14;19(3):e029. b) The total number of alveolar epithelial cells type II (N(AEII, lung)) shows a significant increase after the end of bulk alveolarization and in adults compared to 3 days old rats. c) During alveolarization the number of AEII per mm² alveolar surface (N(AEII, lung)/S(alveoli, lung)) decreases continuously reaching the significant lowest values in adults. d) The number of AEII related to lung volume decreases continuously during alveolarization. The significant lowest values are found in adults. Thus, the lung volume increases faster than the number of AEII in the postnatal development.

## Characterization of Lb

The $V_V$(Lb/AEII) and $S_V$(Lb/AEII) as reference dependent parameters characterized the volume and surface density of Lb in the AEII. While the $V_V$(Lb/AEII) exhibited the highest values at the end of bulk alveolarization, was $S_V$(Lb/AEII) significantly increased in adults. (Fig 8a, 8b). Both parameters correlated well during alveolarization and over all age groups (Table 2, 3).

**Table 2. Regression analysis between different parameters during alveolarization.**

| Relationship between pulmonal variables | Determination coefficient $r^2$ | Correlation coefficient r | Significance (P value) |
|---|---|---|---|
| N(AEII, lung) vs V(lung) | 0.82 | 0.91 | 0.0001 |
| N(AEII,lung) vs S(alveoli, lung) | 0.68 | 0.83 | 0.0001 |
| N(AEII, lung) vs vN(AEII) | 0.05 | 0.22 | n. s. |
| vN(AEII) vs V(lung) | 0.008 | 0,09 | n. s. |
| vN(AEII) vs S(alveoli, lung) | 0.01 | 0.11 | n. s. |
| $V_S$-ratioLb vs $V_S$-ratioAEII | 0.29 | 0.54 | 0.001 |
| V(Lb, lung) vs V(lung) | 0.98 | 0.91 | 0.0001 |
| V(Lb, lung) vs S(alveoli, lung) | 0.71 | 0.92 | 0.0001 |
| S(Lb, lung) vs V(lung) | 0.80 | 0.91 | 0.0001 |
| S(Lb, lung) vs S(alveoli, lung) | 0.75 | 0.8 | 0.0001 |

vs = versus, n.s. = not significant, AEII = alveolar epithelial cells type II, Lb = lamellar bodies

N(AEII, lung) = number of AEII per lung, V(lung) = lung volume,

vN(AEII) = number-weighted mean volume of AEII, S(alveoli, lung) = total alveolar surface area,

$V_S$-ratioAEII = volume to surface ratio of AEII, $V_S$-ratio Lb = volume to surface ratio of Lb,

V(Lb, lung) = total volume of Lb per lung, S(Lb, lung) = total surface of Lb

**Table 3. Regression analysis between different parameters over all age groups.**

| Relationship between pulmonal variables | Determination coefficient $r^2$ | Correlation coefficient r | Significance (P value) |
|---|---|---|---|
| N(AEII, lung) vs V(lung) | 0.96 | 0.98 | 0.0001 |
| N(AEII, lung) vs S(alveoli, lung) | 0.96 | 0.98 | 0.0001 |
| N(AEII, lung) vs v(NAEII, lung) | 0.50 | 0.71 | 0.0001 |
| vN(AEII) vs V(lung) | 0.62 | 0,58 | 0.03 |
| vN(AEII) vs S(alveoli, lung) | 0.58 | 0.76 | 0.0001 |
| $V_S$-ratioLb vs $V_S$-ratioAEII | 0,005 | 0,068 | n. s. |
| V(Lb, lung) vs V(lung) | 0.92 | 0.95 | 0.0001 |
| V(Lb, lung) vs S(alveoli, lung) | 0.93 | 0.95 | 0.0001 |
| S(Lb, lung) vs V(lung) | 0.97 | 0.95 | 0.0001 |
| S(Lb, lung) vs S(alveoli, lung) | 0.94 | 0.95 | 0.0001 |

vs = versus, n.s. = not significant, AEII = alveolar epithelial cells type II, Lb = lamellar bodies

N(AEII, lung) = number of AEII per lung, V(lung) = lung volume,

vN(AEII) = number-weighted mean volume of AEII, S(alveoli, lung) = total alveolar surface area,

$V_S$-ratioAEII = volume to surface ratio of AEII, $V_S$-ratio Lb = volume to surface ratio of Lb,

V(Lb, lung) = total volume of Lb per lung, S(Lb, lung) = total surface of Lb

Determining the volume and surface of Lb per AEII by multiplying $V_V$(Lb/AEII) or $S_V$(Lb/AEII) by vN(AEII), before and during alveolarization no changes were visible. Significant higher values of V(Lb, AEII) and S(Lb, AEII) were determined in adults compared to the postnatal 3 weeks (Fig 8c, 8d).

Multiplying V(Lb, AEII) or S(Lb, AEII) with N(AEII, lung) you get the total volume or surface of Lb per lung (V(Lb, lung)). The V(Lb, lung) was significantly higher at the end of bulk alveolarization compared to values before alveolarization (Fig 8e). The S(Lb,lung) increased significantly at the end of bulk alveolarization. Compared to 3 and 7 days old pups, significant alterations were also found in adults (Fig 8f).

The quotient V(Lb, lung)/$V_{lung}$ as well as the quotient of S(Lb, lung)/$V_{lung}$ did not change before, during and after alveolarization (Fig 9a,b). Thus, Lb volume and lung volume increased proportionally. This was confirmed by a highly significant

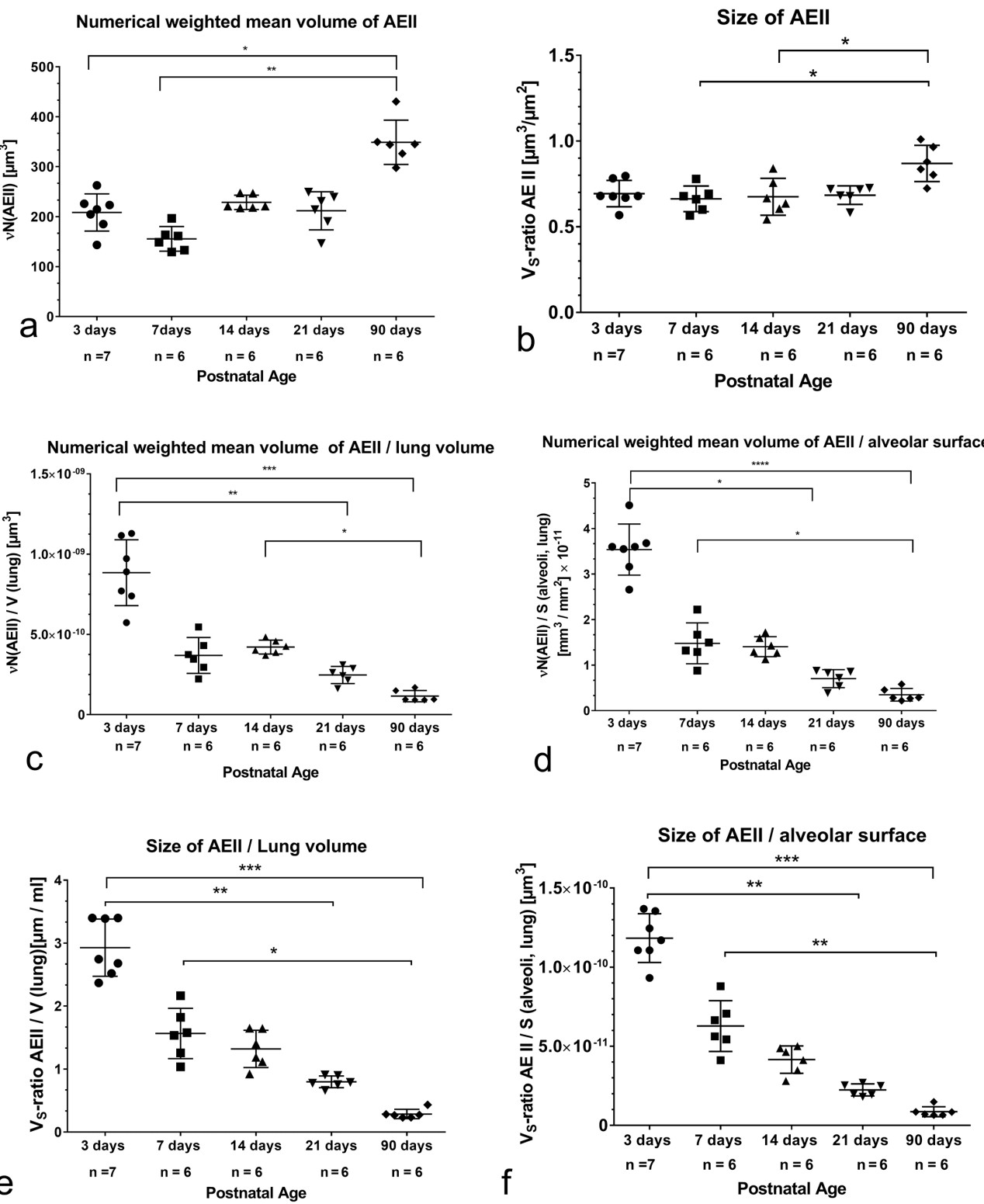

**Fig 5.** a) Number weighted mean volume of AEII during postnatal development and in adults. There are no volume alterations before and at the end of bulk alveolarization. A significant increase was determined in adults compared to 3 and 7 days old pups. b) Size of AEII presented as V_s-ratioAEII during postnatal lung development into adulthood. Before and during bulk alveolarization there are no size changes. Adults show a significant increase

compared to values before and at onset of alveolarization. c) Number weighted mean volume of AEII per lung volume during postnatal development and in adults. The highest values are seen before alveolarization. During alveolarization the quotient decreases and reaches significance compared to 3 days old pups at postnatal day 21. Thus, the lung volume has increased more rapidly than the AEII volume at the end of bulk alveolarization. d) Number weighted mean volume of AEII per alveolar surface area during postnatal development and in adults. Compared to adults, the significantly highest values are seen before alveolarization. e) $V_s$-ratio AEII related to lung volume before and during postnatal development up to adulthood. Lungs in the saccular stage show the highest values compared to 21 days old and adult rats. Therefore, lung volume increases much more than the does AEII increase. f) $V_s$-ratio AEII related to alveolar surface area before and during postnatal development up to adulthood. The significant highest values are seen before alveolarization, the lowest in adults.

correlation between total Lb volume and lung volume (Table 2, 3). Looking at the quotient V(Lb, lung)/S(alveoli, lung) and S(Lb, lung)/S(alveoli, lung), we found a significant decrease at the end of bulk alveolarization compared to values at postnatal day 3 (Fig 9c, 9d). That means that during bulk alveolarization the alveolar surface increased considerably faster than volume or surface of Lb. However, linear regression analyses exhibited a significant correlation between Lb volume or surface with alveolar surface (Table 2, 3).

The volume weighted mean volume of Lb (vV(Lb, AEII)) was comparable during alveolarization and decreased significantly in adults (Fig 9e). The ratio of vV(Lb, AEII) to lung volume showed the highest values before alveolarization, decreased significantly after the end of bulk alveolarization and exhibited the significant lowest values in adults (Fig 9f). Thus, the increase of lung volume was much more pronounced than the decrease in the volume weighted mean volume after alveolarization and in adults.

The size of Lb determined as $V_s$-ratioLb was comparable over all age groups (Table 4). No relationship exists between $V_s$-ratioLb and $V_s$-ratioAEII (Table 2, 3). The $V_v$mvb/AEII and the $V_v$Cb/AEII, the precursors of Lb as well as the energy providing $V_v$(Mi, AEII) showed no alteration comparing values before, during and after alveolarization and in adults (Table 4, 5). However, the V(mito, lung) increased significantly at postnatal day 21, while the V(mi, AEII) remained constant (Table 5).

## Discussion

Our results showed that there is a relationship between growth of AEII as well as their surfactant storing organelles and lung volume as well as alveolar surface before, during and after the end of bulk alveolarization. The increase of alveolar surface area before, during and after alveolarization correlates with the number of AEII as well as with total Lb volume. While alveolar surface area and number of AEII increased significantly at the end of bulk alveolarization, size and volume weighted mean volume of Lb as well as the size distribution and total Lb volume in AEII remained nearly constant. Thus, the AEII has already a morphological mature intracellular surfactant, before alveolarization starts. The adaption to the increasing alveolar surface during animal and lung growth was also followed only by increase of the AEII number. The adaption of the intracellular surfactant pool to the enlargement of the lung primarily by the increase of AEII number and the associated increase in total Lb volume per lung was also found by investigations of lungs of differently sized species [59]. Thus, growth of lungs during development as well as the species dependent lung size is associated with a higher number of AEII and thus a higher total Lb volume per lung to cover the higher need of surfactant.

### Alveolarization

As already shown on the same perfusion fixed lungs, the absolute alveolar surface area as indirect sign of alveolarization, increased significantly by 5.2 times at pnd 21 compared to pnd 3. This increase goes along with formation of smaller secondary septa and change from the double layered to the single layered capillary bed up to the end of bulk alveolarization. The degree of alveolarization during the first 3 weeks is in accordance with older data. Vidic and Burri determined

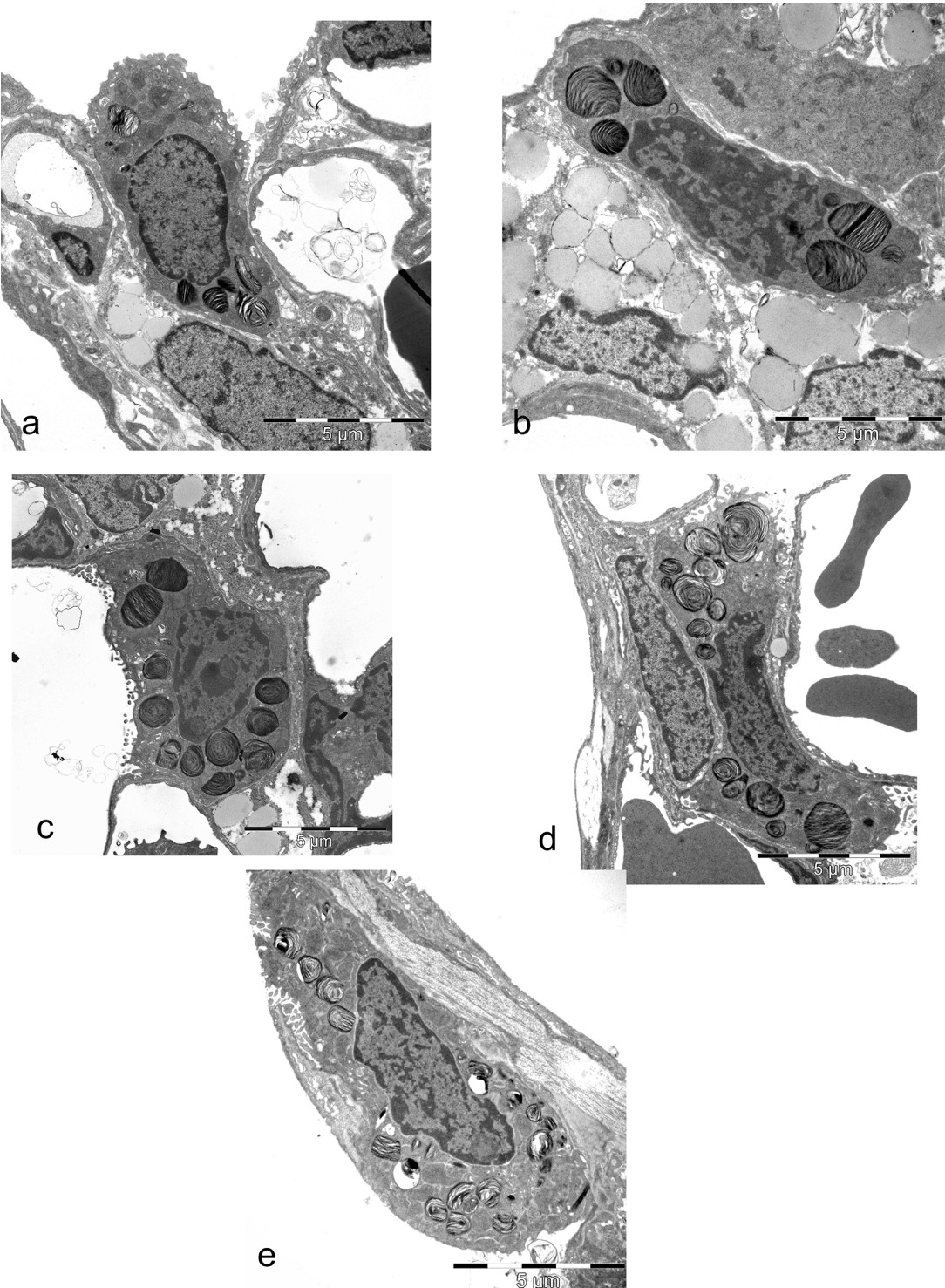

**Fig 6. Representative ultrastructure of AEII from different postnatal development stages.** a) 3 days old: AEII in a corner of the alveolar epithelium between capillaries. The nucleus contains small clumps of heterochromatin on the inner side of the nuclear envelope. In the cytoplasm, organelles such as mitochondria with densely packed cristae and dark matrix, ER, Golgi complex and vesicles of different origin are normally distributed. Normally size distributed Lb are visible. b) 7 days old: AEII below the alveolar surface partly adjacent to lipofibroblasts. The nucleus contains a lot of heterochromatin. Organelles are regularly distributed. Large looking Lb show densely packed lamellae. c) 14 days old: AEII with a visible nucleolus in the nucleus and numerous Lb of different size. d) 21 days old: AEII with nucleus and numerous normally distributed Lb of different size. e) Adult: AEII with normally distributed and preserved organelles.

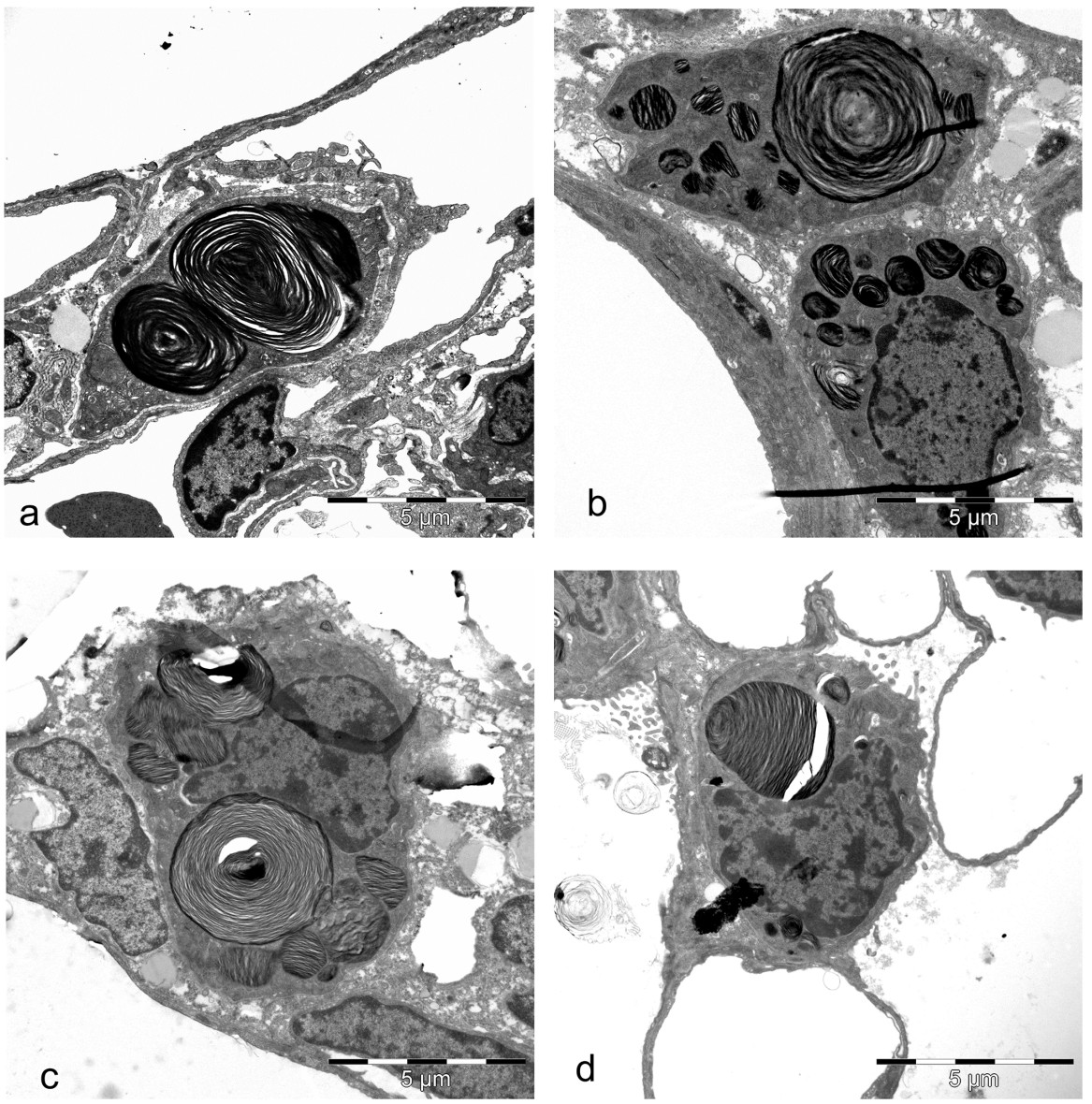

**Fig 7. Ultrastructure of AEII of different stages of development showing giant Lb.** a, b) 3 days old pups, c) 7 days old pups, d) 21 days old rat.

an increase of alveolar surface by 5.6 times from the first to the 21st day of rat life [60]. Already in 1974, the Weibel group presented a comprehensive morphometric study of rat lungs showing an increase of the epithelial and endothelial surface as well as the number of alveoli per section area combined with a decrease in septal thickness during the first 21 pnd [45]. The number of alveoli increased significantly within this time [25,33]. The continued alveolarization phase starts at the end of pnd 21 and lasts at least until adolescens 60 pnd in rats [14,25]. We didn´t investigate this age. In our 90 d old rats (adults), body mass and lung volume as well as all stereological parameters characterizing AEII and Lb exhibit a further increase without reaching significance compared to values obtained to the end of bulk alveolarization, because of the partly high standard deviation in adults as a result of using both sexes.

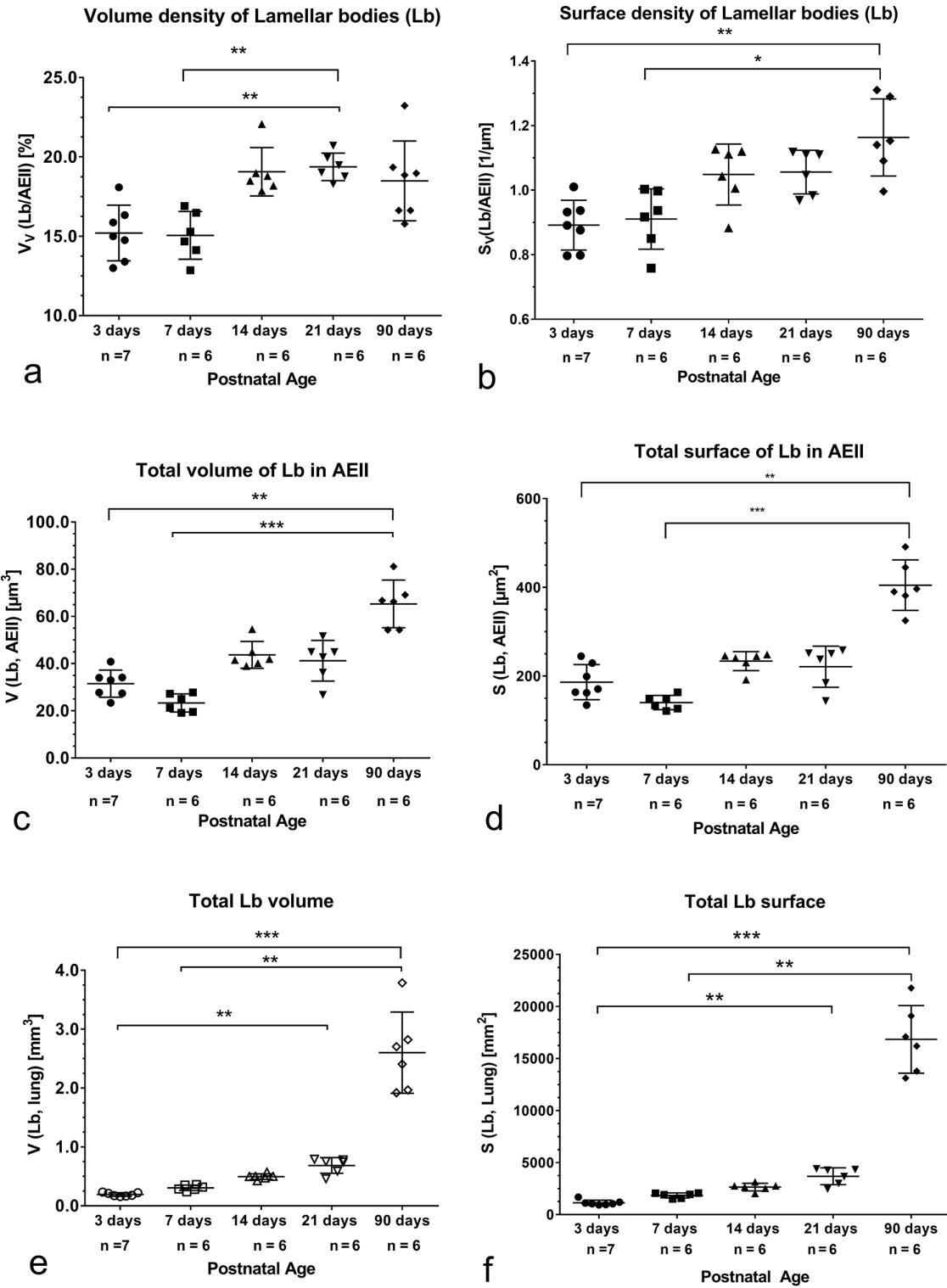

**Fig 8. Various parameters characterizing Lb before, during and after bulk alveolarization (adulthood).** a) Volume density of Lb. In lungs of 21 days old rats significant higher values compared to 3 days and 7 days old rats were determined. b) Before and during alveolarization the surface density of Lb doesn´t change. A significant increase was found in adults compared to 3 and 7 days old rats. c) The total volume of Lb in AEII obtained by multiplying $V_V$(Lb/AEII) with vN(AEII, lung) is comparable before and during bulk alveolarization. A significant increase was determined in AEII of adults. d)

The total surface of Lb in AEII obtained by multiplying $S_v$(Lb/AEII) with vN(AEII) doesn´t change until postnatal day 21. A significant increase compared to 3 and 7 days old lungs was found in adults. e) The total volume of Lb in the lung calculated by multiplying $V_v$(Lb/AEII) with vN(AEII) and N(AEII, lung) over all age groups. Between 3 and 21 days old lungs a significant increase was found. Compared to 3 and 7 days old pups the significant highest values are seen in adults. f) The total surface of Lb in the lung calculated by multiplying $S_v$(Lb/AEII) with vN(AEII) and N(AEII, lung) over all age groups. Between 3 and 21 days old lungs a significant increase was found.

## AEII and alveolarization

Our results are partly in accordance with others using slightly different postnatal times, different experimental design and evaluation methods [48,60,61]. Vicic and Burri showed that the AEII population increased almost its total volume six times, so that the AEII make up 58% of the epithelial volume [60]. They determined a significant increase of the number of AEII on postnatal day 13 compared to day 4 as well as on day 21 compared to day 13 [60]. Randell et al. found no significant alterations in AEII volume, but a significant increase in number within the investigated 14 days compared to the first post-natal day [48]. Kauffman et al determined an increase of the number of AEII up to postnatal day 13, and then a decrease on day 21 [61]. The cell numbers in these papers were determined as number per unit section area, leading possibly to a preferential counting of larger labelled AEII in contrast to our determination using the disector method counting cells independent of size [53]. However, using design-based stereological methods we found no alterations during the first two weeks, but a significant increase an postnatal day 21.

However, determining the number of AEII per mm$^2$ alveolar surface as well as the number of AEII per lung volume, we surprisingly found a significant decrease between postnatal day 7 and 21. Compared to values before alveolarization, the significant lowest values are found in adults (Fig 4c, 4d). That means that at the end of bulk alveolarization and in adults less AEII cover one mm$^2$ of the alveolar surface. Already Randell et al calculated the highest number of AEII per alveolus on the first postnatal day decreasing significantly on day 7 to values, which were comparable with adults [48]. This assessment was supported by determining the ratio of S(alveoli, lung)/N(AEII, lung). The values showed that one AEII may produce surfactant for 1041 μm$^2$ alveolar surface in newborn and in adults for 2534 μm$^2$. During alveolarization the alveolar surface per AEII showed a reduction to 820 μm$^2$ after the first week, then a continuous increase (1441 μm$^2$μm after 14 days and 1835 μm$^2$ after 21 days). This is interesting, because interspecific studies showed that the larger the radius of the alveolar curvature, the higher is a need for surfactant [62], because according to the LaPlace´s law the greater the radius of alveoli, the more surfactant is required. Despite of comparable numbers of AEII during bulk alveolarization and a significant increase at the end of bulk alveolarization on day 21 the number of AEII per alveolus decrease. Thus, less AEII are needed to provide surfactant for the increasing alveolar surface compared to the onset of alveolarization. Therefore, we suggest that the number of AEII, especially the number of AEII per sacculus/alveolus is already sufficient at birth to provide enough surfactant to prevent collapse of the distal airspaces and to help removing alveolar liquid [48]. A call for more surfactant may be accompanied by a higher provision and secretion of surfactant without clearly structural correlates.

The decrease of AEII per alveolar surface during alveolarization may be predominantly caused by the differentiation of AEII into AEI, because AEII are the stem cells for AEI [63], because only stretching of AEI is not sufficient to cover the increased airspace. Using combined morphometric and autoradiography methods it was proofed that only AEII could proliferate after birth, and are therefore responsible for the maintenance of both cell types during alveolarization [61]. Stereological and immunohistochemical methods should prove directly the differentiation of AEII into AEI using double labelling against SP-C protein to see the increase of AEII numbers and markers like Sca-1, which is upregulated during AEII activation and differentiation.

## Intracellular surfactant and alveolarization

The total intracellular volume of pulmonary surfactant, morphologically defined as the total volume of the storage sites, i.e., the Lb [64] is indispensable for normal breathing. Corresponding to the significant increase of AEII number, the total

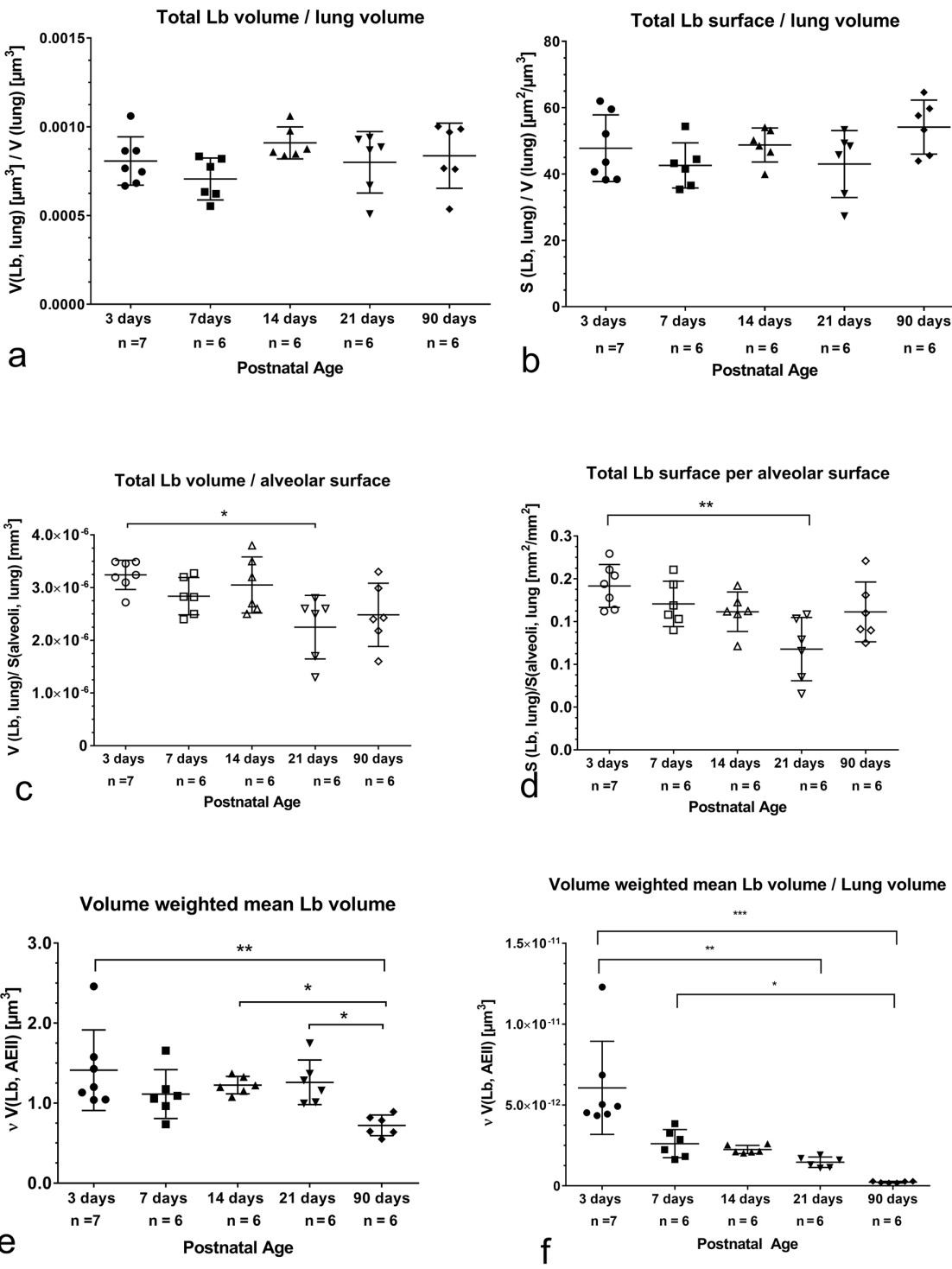

**Fig 9. Various parameters characterizing Lb related to lung volume or alveolar surface before, during and after bulk alveolarization (adulthood).** a) Total volume of Lb related to lung volume. Independent of the age similar values are visible. b) Total surface of Lb related to lung volume. All age groups show similar values. c) Total volume of Lb related to alveolar surface area. During alveolarization values remain constant. At pnd 21 a significant decrease compared to 3 days old pups is visible. Thus, at pnd 21 the increase of alveolar surface is much more pronounced then the increase of Lb volume. d) Total surface of Lb related to alveolar surface area. During alveolarization no differences were found. At pnd 21 values are significantly lower compared to 3 days old pups. Thus, on pnd 21 the increase of alveolar surface is much more pronounced then the increase of Lb surface. e) The

volume weighted mean volume of Lb (vV(Lb, AEII)) exhibits comparable values before, during and after alveolarization. Values determined in adults are significant lower.f) vVLb, AEII) related to lung volume shows the highest values before onset of alveolarization. The significantly lowest values are found in adults. Thus, lung volume increases more rapidly than the volume weighted mean volume of Lb.

**Table 4. Volume densities of subcellular compartments in AEII.**

| Age | $V_V$(cyto,AEII) [%] ** | $V_V$(nucle-i,AEII) [%]** | $V_V$(Lb,AEII) [%] ** | $V_V$composite bodies,AEII) [%] | $V_V$(multives. bodies,AEII) [%] | $V_S$-ratioLb [$\mu m^3/\mu m^2$] |
|---|---|---|---|---|---|---|
| 3 d | 49.90±1.36** | 25.79 ± 1.37 | 15.21±1.75 | 0.56±0.13 | 0.33±0.14 | 0.17±0.02 |
| 7 d | 45.00±1.99 | 30.07±2.70 | 15.07±1.50 | 0.47±0.12 | 0.23±0.12 | 0.17±0.02 |
| 14 d | 44.40±1.40 | 26.53±2.07 | 19.08±1.53 | 0.60±0.06 | 0.37±0.14 | 0.18±0.02 |
| 21 d | 46.32±1.94 | 24.50±2.54 | 19.37±0.86*# | 0.67±0.31 | 0.28±0.12 | 0.19±0.01 |
| 90 d | 50.97±3.01## | 19.37±0.79#+ | 18.80±2.59 | 0.92±0.25 | 0.35±0.15 | 0.17±0.02 |

$V_V$ = volume density, cyto = cytoplasma, AEII = alveolarepithelial cells type II, Lb = lamellar bodies,

$V_S$-ratio = volume to surface ratio, multives, = multivesicular

** There is a signifcant difference between the different age groups (p < 0.0001 (Kruskal-Wallis).

Post hoc test: significant differencens: *with p < 0.05 compared to 3 d, #with p < 0.05 compared to 7 d, +with p < 0.05 compared to 14 d, ##with p < 0.05 compared to 7 ans 14 d, **with p < 0.05 compared to 7d and 14 d

**Table 5. Volume fraction, total volume per AEII and total volume per lung of mitochondria.**

| Postnatal days | $V_V$(mito,AEII) [%] | V(mito,AEII) $\mu m^3$]** | V(mito,lung) $\mu m^3$]** |
|---|---|---|---|
| 3 d | 8.21±0.67 | 17.20±3.87 | 0.092 ±0.045 |
| 7 d | 9.37±1.35 | 13.90±2.50 | 0.182±0.031 |
| 14 d | 9.02 ±1.30 | 20.88±3.13 | 0.238±0.042 |
| 21 d | 8.83 ±0.79 | 18.52±2.58 | 0.309±0.049* |
| 90 d | 9.60 ±1.91 | 37,40±11.44*# | 1.54±0.436*# |

$V_V$ = volume density, mito = mitochondria, AEII = alveolar epithelial cells type II, V = volume,

** There is a signifcant difference between the different age groups (p < 0.0001 (Kruskal-Wallis).

Post hoc test: significant differencens: *with p < 0.05 compared to 3 d, #with p < 0.05 compared to 7 d.

volume and surface of Lb increased also significantly between postnatal day 3 and 21 (Fig 8e-8f), which was also found in an earlier publication [65]. Because of the significant correlation between total Lb volume or surface and alveolar surface from pnd 3 up to the end of bulk alveolarization (pnd 21), we assume that total Lb volume increase is adapted to the increase of alveolar surface and lung volume during alveolarization.

Because increase of total Lb volume and expansion of alveolar surface are constant during the first 14 days (Fig 9c, 9d), we suggest that already three days after birth, before alveolarization starts, the Lb content per cell seems sufficient to reduce surface tension before and during alveolarization. This is consistent with the fact that isolated and cultivated fetal, postnatal and adult rat AEII showed no differences in the content and basal secretion of phosphatidylcholine using sodium [3H-acetat] or [methy-3-H]choline [66]. Thus, despite immature septa with double layered capillary bed and immature saccular terminal air spaces at birth, rats have a morphologically and functionally mature surfactant system. Compared to the alveolarization period, in adults the significantly highest values of total V(Lb, AEII), S(Lb, AEII) as well as V(Lb, lung) and S(Lb, lung), but the lowest vV(Lb, AEII) were evaluated. The size of Lb ($V_S$-ratioLb) was comparable to sizes during alveolarization (Table 4). Therefore, we suggest that the adaption of Lb to the increased lung volume and alveolar surface occurred by the increase of Lb number. As a result, we determined a greater total cellular volume of smaller sized Lb.

Not only AEII and Lb have to be well developed already before alveolarization, but also the surfactant proteins, which were not investigated in this study, have to be sufficiently expressed. It is known that the surfactant protein (SP) mRNA for the SP-A, SP-B as well as SP-C occurred already in the prenatal pseudoglandular phase at gestation day 13 in the distal ducts [43] and that the synthesis of surfactant phospholipids started in the canalicular phase [44]. Own immuno-electron microscopic and stereologic studies proved that in newborn, 14 days old and adult rats the preferential distribution of all four SP was comparable [67]. We also demonstrated that during bulk and continued alveolarization the test fields with active surfactant subtypes increased continuously to guarantee enough surfactant lining the alveoli to prevent collapse of the airspaces and facilitate inflation [47]. Thus, the increase of total Lb volume and the increase of active components of intraalveolar surfactant during alveolarization are an indirect evidence for a higher surfactant secretion. Using molecular biological methods, immunohistochemistry and stereology, we already demonstrated that the expression of SP-A and SP-B is differentially associated with morphological lung development and correlates with the increase of the alveolar surface [41]. SP-A gene expression and the relative surface fraction of SP-A labelled AEII showed the highest values in the first days after birth, which is explainable with its immunological function [68,69] as the animals get in touch with many different pathogens after birth. SP-B gene expression and the fraction of SP-B labelled AEII increased with alveolarization and showed the highest values in adults to guarantee a continuous spreading of surfactant over the increasing inner alveolar surface [41], because SP-B is required for the formation of the surfactant film from the precursors [16].

Summarizing all data it becomes clear that in contrast to human, there is a complete morphological and molecular biological maturation of the surfactant system already in the saccular phase in rodents. Preterm infants born between gestation week 24 and 36 are also in the saccular phase. First surfactant components were yet synthesized and secreted as in rats in the canalicular phase of prenatal human lung development [21]. Surfactant gains first sufficient functional properties in the saccular phase between the 29th and 32nd week of pregnancy in humans [21]. In the late saccular phase, from the 32nd week on, the number of Lb increases. Large quantities of surfactant are present in fetal lungs and amniotic fluid. At this point of lung maturation, the concentrations of lecithin and sphingomyelin are relatively equal [70]. However, infants born before the gestation week 35 do not have enough functional surfactant to breath without any problems. Surface active surfactant components were secreted from the 36th week onwards, and thus with onset of alveolarization [21,22,70]. At this time, a marked increase in the concentration of lecithin in fetal lungs and amniotic fluid was measured leading to a crescent fully functional surfactant. A lecithin-to-sphingomyelin ratio higher than 2:1 is characteristic of mature fetal lungs [70]. A comparison of lung development between rodents and humans reveals that in both, surfactant maturation is finished towards the end of the saccular phase. However, rodents are born in this phase and alveolarization starts postnatal, while in human alveolarization begins and continues prenatal until birth at week 40 p.m.. Preterm infants born in the saccular phase before week 35 have increasing problems with breathing the earlier the birth is, because of the increasing immaturity of the surfactant system. In summary, functionally active surfactant is already synthesized, and secreted in rodents during the saccular phase and in humans during the alveolar phase.

Not least, the provision and secretion of surfactant needs energy. Therefore, we looked also to the mitochondria in AEII (Table 5). As shown for Lb volume in AEII, also the mitochondrial volume in AEII remain constant during alveolarization, while the V(mito, lung) increased significantly at pnd 21, the end of bulk alveolarization as a result of the increase of AEII number. Vidic and Burri found the greatest changes of total mitochondrial volume at the end of bulk alveolarization, too [65]. Thus, 3 days after birth not only Lb volume but also mitochondrial volume per AEII is sufficient so that it remains constant during alveolarization. This assumption is in accordance with biochemical studies showing that before and during alveolarization the basal rate of phosphatidylcholine secretion did not differ. However, the response to secretagogues changed during postnatal lung development [66], which is not morphologically measureable.

## Giant Lb

During scanning ultrathin sections using the systematic random sampling method we found occasionally so called giant Lb not only before, but also during alveolarization (Fig 7a-7d). The occurrence of giant Lb contributes to the amount of variation of $V_V$(Lb/AEII), $\nu$V(Lb, AEII)), Lb size and volume. Because of the rather rare occurring enlargement of Lb, we assume that lung function is not affected. More frequently occurring AEII with such huge Lb combined with clinical symptoms are found in the literature. E.g. in a model of respiratory distress syndrome using rat lungs after intraperintoneal application of LPS [71], in a mice model of Chédiak-Higashi syndrome (CHS), a rare severe genetic disorder generally characterized by partial oculocutaneous albinisma [72], in patients suffering from Hermansky-Pudlak syndrome (HPS) with interstitial pneumonia [73]. We suggest that the occurrence of giant Lb during lung development may be a result of problems to secrete Lb or of cellular disturbances of the phospholipid accumulation. The resulted intracytoplasmic pooling of phospholipids lead then to an enlargement of Lb. A fusion of Lb is also conceivable [71,73]. Because Lb are regarded as lysosome-related organelles [74] it was assumed that the enlargement of Lb may be the result of a defect of phospholipase activity [73]. Phospholipase A2 activity is important for phospholipid synthesis in AEII and is regarded as a mechanism for enriching DSPC within lamellar body surfactant. Further investigations showed that AEII lacking the adaptor protein 3 complex (AP-3) fails to accumulate the soluble enzyme peroxiredoxin a primary phospholipase A2 in the lumen of Lb resulting in an enlargement of Lb using a pearl mouse model of HPS type 2 [75]. More studies are necessary to explain the occurrence of giant Lb during alveolarization.

## Limitations

This study has a few limitations, which are discussed below.

**Lung function.** This study focused predominantly on the morphological development of AEII before, during and after alveolarization. Additional lung function studies for each postnatal age should give more insights into changes of the link between respiratory function and lung structure. Thereby the question arises whether changes in AEII influence lung function. Looking in the literature, some investigations of the link between lung function and lung structure regarding the alveolar surface and airspace volume were done [76–78]. Using a computer controlled animal ventilator, Gomes and coworkers measured respiratory mechanics (mechanical resistance and elastance) at different levels of positive end-expiratoy pressure (PEEP) from early age to adulthood in living anesthetized rats [76]. The investigated ages (10, 14, 18, 21, 25, 30, 90 days) are partly comparable with the ages of our investigated rats (3, 7, 14, 21, 90 days). Generally, they found that airway resistance as well as elastance or stiffness defined as reciprocal compliance decrease considerably with age because of animal growth and increase in lung size. Furthermore, they showed higher strain stiffening of lung tissue at lower distending pressures during alveolarization than after alveolarization and suggested that stiffer lung tissue results in greater changes of elastic recoil forces for a given change in lung volume [76]. Therefore, increased interdependence forces occur between airway and lung parenchyma. The uncoupling between airways and parenchymal tissue and therefore the mechanical interdependence between airway and parenchyma is weaker in young animals than in mature animals leading to a hyperresponsiveness of immature lungs [76]. Similar results were found in mice. Here, also airway resistance decrease with increasing alveolarization and tissue damping and tissue elastance decreased beyond 2 weeks until 5 weeks [79] Thus, the mechanical changes alter during morphologic lung maturation, and are linked with a decrease in mean tissue density and increase in total alveolar surface area. According to their results, our findings regarding the AEII number and Lb volume may have a negligible influence on lung mechanics. In another study, in addition to respiratory mechanics, total lung capacity (TCL, lung volume), dead space volume ($V_D$), lung volume, gas exchange and intrapulmonary gas mixing were examined as well as structural parameters (lung alveolar surface area, alveolar size and alveolar septal thickness). The evaluations were made during bulk alveolarization (7d, 14d), at the end of bulk alveolarization (21d), during continued alveolarization (35d) and in adults (90d). However, the developmental stage before alveolarization was not investigated [78]. This comprehensive study showed that compared to 7 d old rats significant

alterations of values of functional lung parameters such as total lung capacity, dead space volume static compliance pulmonary diffusion capacity occurred earliest at the end of bulk alveolarization. Furthermore, the static and dynamic compliance increased proportionally with age. The resting expiratory position (FRC/TLC) and the specific residual volume declined with age, because of improved mechanical interdependence between airways and parenchyma [78] as also described by Gomes et al. [76]. The increase of the gas diffusion capacity is closely linked with the increase in alveolar surface area and decrease of septal thickness and depends therefore on morphological maturation of the gas-exchange region, which is finished at the end of bulk alveolarization. So during alveolarization, the series death space volume increased significantly, while the total lung capacity significantly increased, because parenchymal growth is faster than growth of the conducting airways [78]. The increased diffusion capacity is regarded as result of the substantial increase of surface area during alveolarization [78]. These studies showed that provided that the surfactant system is functionally intact to prevent a respiratory distress syndrome, lung functional parameters are in close relation to the development of the alveoli and the gas exchange region. Comparing rats and humans regarding lung development, there are a lot morphological similarities. However, lung function is not readily transferable on a one-to-one basis from rats to humans, particularly during continued alveolarization [78]. E.g., during the development of lung volume from birth to adulthood, the quotient FRC/TLC and therefore the resting expiratory position increases in humans, while there was a decrease in rats [78]. Furthermore, the specific lung size in humans is twice as large as in rats [78]. While, in rats lung volume growth surpasses airway growth, in humans both develop in proportion [80]. Additionally, the specific airway conductance is constant during lung development, while it increases in rats [78]. In summary, lung functional parameters correlate predominantly with the development of the alveolar surface and the gas-exchange region. The maturation of AEII plays a role with regard to sufficient surfactant to enable breathing movements.

**Sex differences.** We did not evaluate separately female and male lungs. Some studies indicate a critical role of sex hormones in fetal lung development. In fetal rat and rabbit formation of Lb and increase of AEII was stimulated by estrogens [81]. Independent of the species a delay in lung maturation and therefore surfactant production was found at the transition from the prenatal canalicular into the saccular stage in males [82,83]. However, no sex differences were found in the very immature or the mature rabbit fetuses [84]. Furthermore, no sex-related differences of different parameters (body weight, lung weight, right lung volume, lung tissue and airspace fractions, mean linear intercept, septal crest density, septal thickness, proportion of proliferating and apoptotic cells, percentages of collagen, elastin gene expression of surfactant proteins and tropoelastin) were found in lambs 0.9 of term [85]. Thus maybe that the male disadvantage is limited to the late canalicular and early saccular period and plays therefore in our studies a minor role. However an interesting study on mice showed that between 2 and 13 weeks developmental changes of in functional parameters such as inspiratory capacity, lung compliance and hysteresis as well as structural parameters such as mean chord length, lung volume and gas exchange area were weight-dependent rather than sex – dependent [77,86]. Therefore, further studies should be carried out regarding the sexes.

### Litter size and environmental conditions

We used rat pups from up to three litters. The dams gave birth to litters with 6–12 newborn pups. The environmental conditions were the same for each dam and all pups.

Litter size may effect growth and development of pups starting already during prenatal development. Large litter sizes correlate with limited milk yield and impaired maternal behavior [87]. Furthermore, an inverse relationship between pup body weight and litter size during the whole suckling period until weaning at postnatal day 21 as well as a delay in maturational milestones (fur development, incisor eruption, eye opening), but the occurrence of pup death is independent of the litter size [87]. Though, using design based stereology the research group of Morty concluded that alveolarization is not delayed in mouse pups selected from litters between 2 and 8 [88]. We assume that this also applies to rats. Additionally, in our study, we selected pups exhibited similar trends in body mass range. Because we used pups selected only from

litters of 1–3 dams, a possible not sufficient variability between pups of different dams should be regarded. However, to our knowledge there are no studies indicating an influence on the data comparing pups of litters mainly from one or more dams. The advantage of our pup selection is that no coiling of surplus of unused pups was necessary. Looking at the literature dealing with lung development, information on the number of litters from which pups of one age were used, is rare. Either no information is provided [35,48,77], or no sound information was given. E. g. "young rats from about seven litters of the CFN-COBS-strain were killed in groups of three at the age of 1, 4, 7, 10, 13 and 21 days" [45], respectively, "at each of postnatal days 4, 10, 21, 36, and 60, five animals were taken for lung fixation" [25]. This should be taken into account when results are considered and classified.

## Conclusion

Thus, rats are born with morphologically immature lungs, but the AEII have already sufficient mature surfactant storage sites to produce enough surfactant allowing breathing and preventing postnatal collapse of the gas exchange region. The surfactant system is adapted to alveolarization predominantly by increase of the AEII number. The volumes and sizes of the AEII and the cellular volumes and sizes of Lb within the AEII cells do not vary very much.

## Acknowledgements

We thank Susanne Faßbender, Andrea Herden and Sabine Fiedler for their expert technical assistance. Part of the work is the medical thesis of Julia Hüttmann.

## Author contributions

**Conceptualization:** Andreas Schmiedl.

**Data curation:** Andreas Schmiedl, Julia Hüttmann.

**Formal analysis:** Andreas Schmiedl, Julia Hüttmann, Lars Knudsen.

**Investigation:** Andreas Schmiedl, Julia Hüttmann.

**Methodology:** Andreas Schmiedl, Lars Knudsen.

**Project administration:** Andreas Schmiedl.

**Supervision:** Andreas Schmiedl.

**Validation:** Andreas Schmiedl, Julia Hüttmann, Lars Knudsen.

**Visualization:** Andreas Schmiedl, Julia Hüttmann, Lars Knudsen.

**Writing – original draft:** Andreas Schmiedl, Julia Hüttmann.

**Writing – review & editing:** Andreas Schmiedl, Julia Hüttmann, Lars Knudsen.

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
