## [Decision Letter · Decision Letter 0]

15 Jul 2025

PONE-D-25-27042Characterization of alveolar epithelial cells type II during postnatal lung development in relation to alveolarization – Stereological studies of rat lungsPLOS ONE

Dear Dr. Schmiedl,

Thank you for submitting your manuscript to PLOS ONE. After careful consideration, we feel that it has merit but does not fully meet PLOS ONE’s publication criteria as it currently stands. Therefore, we invite you to submit a revised version of the manuscript that addresses the points raised during the review process.

We look forward to receiving your revised manuscript.

Kind regards,

Zissis C. Chroneos, Ph.D.

Academic Editor

PLOS ONE

Journal Requirements: 

2. To comply with PLOS ONE submissions requirements, in your Methods section, please provide additional information regarding the experiments involving animals and ensure you have included details on methods of sacrifice, and efforts to alleviate suffering.

3.  We note that Figure 2 a in your submission contain copyrighted images. All PLOS content is published under the Creative Commons Attribution License (CC BY 4.0), which means that the manuscript, images, and Supporting Information files will be freely available online, and any third party is permitted to access, download, copy, distribute, and use these materials in any way, even commercially, with proper attribution. For more information, see our copyright guidelines: http://journals.plos.org/plosone/s/licenses-and-copyright.

1. You may seek permission from the original copyright holder of Figure 2 a to publish the content specifically under the CC BY 4.0 license.

Reviewers' comments:

Reviewer's Responses to Questions

**Comments to the Author**

1. Is the manuscript technically sound, and do the data support the conclusions?

Reviewer #1: Yes

Reviewer #2: Yes

2. Has the statistical analysis been performed appropriately and rigorously? 

Reviewer #1: No

Reviewer #2: Yes

3. Have the authors made all data underlying the findings in their manuscript fully available?

Reviewer #1: Yes

Reviewer #2: Yes

4. Is the manuscript presented in an intelligible fashion and written in standard English?

Reviewer #1: No

Reviewer #2: Yes

5. Review Comments to the Author

Reviewer #1: I thank authors for giving me an opportunity to review this interesting work. In this submitted paper, the authors have studied the relationship between growth of alveolar epithelial cells type 2 and their surfactant storing organelles to alveolar surface and lung volume before, during and after the end of bulk alveolarization in comparison to adults using the stereological methods. Having said that the authors need to clarify few things to improve the manuscript.

Major comments:

1. Table 1 is unclear, particularly it is hard to believe that there is not statistical difference between weight and lung volume between different ages. One-way ANOVA test would be an appropriate analysis for this. For example, the weight difference between 3- and 7-days old pups are almost twice (6.47+-0.39 g vs 12.6+-0.34 g) with not a major standard deviation, yet it is not significantly different. The same goes for the lung volume.

2. Table 2. The p value is given with percentage (%). The authors can remove that part.

3. The authors should consider additional experiments to support their findings of increased alveolar epithelial cell type 2 cells as they mature and also need additional experiment to prove their hypothesis that AE2 differentiate into AE1 as the rat ages. The markers to look for are SP-C protein, SFTPC gene expression to see increase AE2 cells numbers. Moreover, markers like Sca-1, which is upregulated during AT2 cell activation and differentiation, can help track this process (AE2 to AE1 differentiation).

4. In addition, the authors could also consider performing lung function tests to study the impact of these changes on physiological parameters and lung function as rat ages.

5. The authors need to improve the discussion part. There is a lot of repetition of results rather authors should focus on central and important results and what that means in terms of previous literature and how it adds to the new literature. Moreover, the authors also need to expand on how these findings relate to humans. How does it correlate with humans? I recommend authors to expand a discussion that would enhance utility and relatability of the study and how rat model would be an ideal model to study these changes (the last point could be included in the introduction).

Minor comments:

1. Lots of punctuation errors which hinders the flow of the article, particularly the references are not cited properly at many places.

2. The authors could consider using the different colors for arrows for more clarity in the figure.

Reviewer #2: 1. The study addresses a significant gap in the field of developmental lung biology. Rats are born with morphologically immature lungs, yet they possess a functional surfactant system. However, the relationship between surfactant-producing alveolar epithelial type II cells and the process of alveolarization remains inadequately understood. Understanding this relationship is essential for gaining insights into lung maturation and the potential development of neonatal respiratory disorders.

2. Authors offer a substantial morphological and quantitative framework for comprehending AEII cell dynamics during postnatal lung development. It demonstrates that AEII cells primarily adjust to the increasing alveolar surface area through proliferation rather than hypertrophy, while also establishing their surfactant storage capacity at an early stage.

3. The results section regarding the body weight and lung volume of rats would benefit from more comprehensive statistical reporting. In reference to Table 1, it should be noted that body mass and lung volume changed during the first two postnatal weeks, contradicting the initial assertion that "the body weight and lung volume did not change during the first two postnatal weeks." Additionally, Tables 2, 3, 4, and 5 lack legends, which should include abbreviations and significance data for clarity.

4. The functional implications of the study findings remain speculative, without physiological or molecular validation. In the limitations section, authors are recommended to address the following points:

a) Absence of functional data, specifically a direct assessment of surfactant production, secretion, or gas exchange efficiency.

b) Lack of gene or protein expression data that could elucidate the underlying molecular mechanisms.

c) For example- Programmed cell death was referenced, but it was not quantitatively measured in this study.

d) Confounding factors, aside from sex, such as litter size and environmental conditions, including nutrition, have not been disclosed.

6. PLOS authors have the option to publish the peer review history of their article (what does this mean? ). If published, this will include your full peer review and any attached files.

**Do you want your identity to be public for this peer review?** For information about this choice, including consent withdrawal, please see our Privacy Policy .

Reviewer #1: No

Reviewer #2: **Yes: ** Shaili Amatya

---

## [Author Response · Author response to Decision Letter 1]

17 Oct 2025

Responses to the comments of the editor and reviewers

We wish to thank you and the reviewers for your constructive and valuable reviews and feedback, and for the opportunity to submit a revised version of the manuscript.

The point by point responses to all the comments by the editor and reviewers are given below.

Editor

To comply with PLOS ONE submissions requirements, in your Methods section, please provide additional information regarding the experiments involving animals and ensure you have included details on methods of sacrifice, and efforts to alleviate suffering.

Response: Yes, we did.

We require you to either (1) present written permission from the copyright holder to publish these figures specifically under the CC BY 4.0 license, or (2) remove the figures from your submission

Response: Because the image in question originates with PLOS ONE and was thus initially published under the CC BY 4.0 license, written permission is not needed to reproduce this image.

Response: There is no permission necessary. We uploaded the letter from the Journal.

We indicate in the Figure legend the origin of the figure and that the original Figure image has been altered (p. 27, lines 4-6).

Reviewer 1: Major comments:

1. Table 1 is unclear, particularly it is hard to believe that there is not statistical difference between weight and lung volume between different ages. One-way ANOVA test would be an appropriate analysis for this. For example, the weight difference between 3- and 7-days old pups are almost twice (6.47+-0.39 g vs 12.6+-0.34 g) with not a major standard deviation, yet it is not significantly different. The same goes for the lung volume.

Response: We added the calculated statistical values. We used not the One-Way ANOVA test for normally distributed values, but the nonparametric Kruskal-Wallis test for not normally distributed values to get an information whether there are differences of the values between the age groups. We added the post-hoc Dunn´s test to adjust the p-values for multiple comparisons of differentially aged groups. This test is recommended, because the Mann-Whitney-Wilcoxon test uses ranks of only two groups at a time. That’s different from the Kruskal-Wallis test statistic, which calculates ranks shared across all the group and the same data were used for the two-way comparison. Due to the higher apha-level, only a few significances were found between the age groups. Using the Mann-Whitney test additional to the Kruskal-Wallis test, an adjustment like Bonferroni is also necessary, which also limits the significances

2. Table 2. The p value is given with percentage (%). The authors can remove that part.

Response: We removed the percent sign.

3. The authors should consider additional experiments to support their findings of increased alveolar epithelial cell type 2 cells as they mature and also need additional experiment to prove their hypothesis that AE2 differentiate into AE1 as the rat ages. The markers to look for are SP-C protein, SFTPC gene expression to see increase AE2 cells numbers. Moreover, markers like Sca-1, which is upregulated during AT2 cell activation and differentiation, can help track this process (AE2 to AE1 differentiation).

Response: Thank you very much for your suggestions. Using double labelling against SP-C and Sca-1, which is upregulated during AEII activation and differentiation helps indeed to get specific information about the AEII into AEI differentiation process. However, a precondition for doing immunohistochemistry is the correct fixation and embedding of samples. We used formaldehyde and glutaraldehyde in our fixation solution to ensure good structural preservation and embedded the specimen in epon for transmission electron microscopy. Thus, the antigenicity of the proteins has been lost. Therefore, for co-staining of SFTPC and Sca-1 we would have to carry out new animal experiments. The best way to retain antigenicity is to instill the lungs with a mixture of cryogel tissue tec OCT/PBS and then freeze them on dry ice. This procedure preserves antigenicity and allows immunohistochemical studies on the light microscopical level. New experiments require a new animal testing application and approval. This would last too long.

Therefore, we uptake the suggestion in the discussion (p. 16, line 17-20).

Not least, using autoradiography Kauffman et al proved the differentiation of AEII in AEII (Kauffman SL, Burri PH, Weibel ER. The postnatal growth of the rat lung. II. Autoradiography. Anat Rec. 1974 Sep;180(1):63-76).

4. In addition, the authors could also consider performing lung function tests to study the impact of these changes on physiological parameters and lung function as rat ages.

Response: That is a very good suggestion but requires a new animal testing application and approval. Therefore, we discuss our results regarding lung function parameters obtained from the literature. Some authors already carried out such experiments in rats of different postnatal age. We included a new chapter “ lung function” (p. 22, lines 6-27, p.23, lines 1-28, p. 24, lines 1-2)

5. The authors need to improve the discussion part. There is a lot of repetition of results rather authors should focus on central and important results and what that means in terms of previous literature and how it adds to the new literature.

Response: We eliminated the redundant sections of our results and focus more on our results in relation to the literature. Furthermore, we eliminated the section “AEII and alveolar surface and their relationships” in the discussion (p. 15, lines 17-21, p. 17, lines 6-16, p.18, lines 3-11, p. 21, lines 1-28, p. 22, lines 1-2) and insert the non-redundant information of this chapter into the other sections

Moreover, the authors also need to expand on how these findings relate to humans. How does it correlate with humans? I recommend authors to expand a discussion that would enhance utility and relatability of the study and how rat model would be an ideal model to study these changes (the last point could be included in the introduction).

Response: We considered the suggestions in the Introduction and Discussion (p. 3, lines 24-27, p. 19, lines 3-22)

Minor comments:

1. Lots of punctuation errors which hinders the flow of the article, particularly the references are not cited properly at many places.

Response: We corrected the punctuation errors and added missing references.

2. The authors could consider using the different colors for arrows for more clarity in the figure.

We used different colors for the arrows.

Reviewer #2:

3. The results section regarding the body weight and lung volume of rats would benefit from more comprehensive statistical reporting. In reference to Table 1, it should be noted that body mass and lung volume changed during the first two postnatal weeks, contradicting the initial assertion that "the body weight and lung volume did not change during the first two postnatal weeks."

Response: We added the calculated statistical values. We used not the One-Way ANOVA test for normally distributed values, but the nonparametric Kruskal-Wallis test for not normally distributed values to get an information whether there are differences of the values between the age groups. We added the post-hoc Dunn´s test to adjust the p-values for multiple comparisons of differentially aged groups. This test is recommended , because the Mann-Whitney-Wilcoxon test uses ranks of only two groups at a time. That’s different from the Kruskal-Wallis test statistic, which calculates ranks shared across all the group and the same data were used for the towo-way comparison. .Due to the higher apha-level, only a few significances were found between the age groups. Using the Mann-Whitney test additional to the Kruskal-Wallis test, an adjustment like Bonferroni is also necessary, which also limits the significances.

Additionally, Tables 2, 3, 4, and 5 lack legends, which should include abbreviations and significance data for clarity.

Response: We added the legends.

4. The functional implications of the study findings remain speculative, without physiological or molecular validation. In the limitations section, authors recommended to address the following points

a) Absence of functional data, specifically a direct assessment of surfactant production, secretion, or gas exchange efficiency.

Response: We discuss this in combination with additional papers in the limitations section p. 22, lines 6-27, p. 23, lines 1-28, p. 24, lines 1-2).

b) Lack of gene or protein expression data that could elucidate the underlying molecular mechanisms.

Response: We discuss this and add some papers dealing with SP expression during postnatal development (p. 18, lines 12-28, p. 19, lines 1-2).

c) For example- Programmed cell death was referenced, but it was not quantitatively measured in this study.

Response: For this purpose, another fixation design would have been necessary. The used fixation solution preserve the fine structure of lung parenchyma well, but permits no immunohistochemical methods, because the antigens lost immunoreactivity. We removed this part from the discussion, because in the cited paper apoptosis was proofed only with the TUNEL method without success to identify characteristic apoptotic signs in AEII using electron microscopy. The authors suggested that these cells were eliminated by macrophages (Schittny JC, Djonov V, Fine A, Burri PH. Programmed cell death contributes to postnatal lung development. Am J Respir Cell Mol Biol. 1998 Jun;18(6):786-93). Furthermore, the TUNEL-method, is not considered to be a specific marker of apoptosis (Sutherland LM, Edwards YS, Murray AW. Alveolar type II cell apoptosis. Comp Biochem Physiol A Mol Integr Physiol. 2001 May;129(1):267-85).

We removed the apoptosis section (p.16, lines 21-24)

d) Confounding factors, aside from sex, such as litter size and environmental conditions, including nutrition, have not been disclosed.

Response: We conclude litter size and environmental conditions in our limitation section (p. 24, lines 21-26, p. 25, lines 1-15)

---

## [Decision Letter · Decision Letter 1]

4 Nov 2025

Characterization of alveolar epithelial cells type II during postnatal lung development in relation to alveolarization – Stereological studies of rat lungs

PONE-D-25-27042R1

Dear Dr. Schmiedl,

We’re pleased to inform you that your manuscript has been judged scientifically suitable for publication and will be formally accepted for publication once it meets all outstanding technical requirements.

Kind regards,

Zissis C. Chroneos, Ph.D.

Academic Editor

PLOS ONE

Additional Editor Comments (optional):

Reviewers' comments:

Reviewer's Responses to Questions

**Comments to the Author**

1. If the authors have adequately addressed your comments raised in a previous round of review and you feel that this manuscript is now acceptable for publication, you may indicate that here to bypass the “Comments to the Author” section, enter your conflict of interest statement in the “Confidential to Editor” section, and submit your "Accept" recommendation.

Reviewer #1: All comments have been addressed

Reviewer #2: All comments have been addressed

2. Is the manuscript technically sound, and do the data support the conclusions?

Reviewer #1: Yes

Reviewer #2: Yes

3. Has the statistical analysis been performed appropriately and rigorously? 

Reviewer #1: Yes

Reviewer #2: Yes

4. Have the authors made all data underlying the findings in their manuscript fully available?

Reviewer #1: Yes

Reviewer #2: Yes

5. Is the manuscript presented in an intelligible fashion and written in standard English?

Reviewer #1: Yes

Reviewer #2: Yes

6. Review Comments to the Author

Reviewer #1: (No Response)

Reviewer #2: The concerns have been addressed, and the authors have replied to the comments. No further questions or comments

7. PLOS authors have the option to publish the peer review history of their article (what does this mean? ). If published, this will include your full peer review and any attached files.

**Do you want your identity to be public for this peer review?** For information about this choice, including consent withdrawal, please see our Privacy Policy .

Reviewer #1: No

Reviewer #2: No

---

## [Editor Report · Acceptance letter]

PONE-D-25-27042R1

PLOS ONE

Dear Dr. Schmiedl,

I'm pleased to inform you that your manuscript has been deemed suitable for publication in PLOS ONE. Congratulations! Your manuscript is now being handed over to our production team.

Kind regards,

on behalf of

Dr. Zissis C. Chroneos

Academic Editor

PLOS ONE